# Choroidal and Choriocapillaris Morphology in Pan-FGFR Inhibitor-Associated Retinopathy: A Case Report

**DOI:** 10.3390/diagnostics12020249

**Published:** 2022-01-20

**Authors:** Giuseppe Fasolino, Laura Moschetta, Jacques De Grève, Pieter Nelis, Pierre Lefesvre, Marcel Ten Tusscher

**Affiliations:** Universitair Ziekenhuis Brussel, Brussels Health Campus, Vrije Universiteit Brussel, Laarbeeklaan 101, 1090 Jette, Belgium; giuseppe.fasolino@uzbrussel.be (G.F.); jacques.degreve@uzbrussel.be (J.D.G.); pieter.nelis@uzbrussel.be (P.N.); pierre.lefesvre@uzbrussel.be (P.L.); marcel.tentusscher@uzbrussel.be (M.T.T.)

**Keywords:** choroidal and choriocapillaris morphology, pan-FGFR Inhibitor-Associated Retinopathy, OCT-angiography, retinal serous detachment, pachychoroid spectrum disease

## Abstract

Emerging anticancer agents such as the pan-FGFR Inhibitor have achieved remarkable improvements in the survival of patients with metastatic malignancies. Nevertheless they are still associated with specific ophthalmic toxicities. Understanding their pathophysiology can lead us to better clinical practice of life-threatening and vision-threatening circumstances. To investigate choroidal alterations as a potential pathophysiological mechanism of a serous detachment in bilateral pan-FGFR Inhibitor-Associated Retinopathy (FGFRAR), the morphology of the choroid and choriocapillaris were assessed. The choroidal thickness (ChT) and choriocapillaris flow void were measured by macular optical coherence tomography (OCT) and angiography (OCT-A), respectively. Data were collected at the baseline, then at one-month and two-months follow-ups after starting erdafitinib, in a single case of pulmonary angiosarcoma. Choroidal and choriocapillaris morphology showed stable ChT and choriocapillaris flow void at FGFRAR onset and relapse. To the best of our knowledge, this is the first analyzed case reported with flow-void OCT-angiography. Considering these results, FGFRAR in this patient does not seem to match the pachychoroid spectrum disorder definition; rather, an intracellular mechanism based on intracellular transduction pathways may be at work.

## 1. Introduction

Over the last two decades, a remarkable improvement in the survival of patients with metastatic malignancies has been achieved. Anticancer agents target tumor-specific genomic alterations that activate signal transduction pathways on which the cancer cells are dependent. In contrast to traditional chemotherapy, targeted agents have a higher selectivity for the cancer cells and a better general tolerance. Nevertheless, many of these agents are still associated with specific toxicities, also at the ophthalmic level.

Most recently, the first pan-fibroblast growth factor receptor inhibitor (FGFRi) has been approved, with similar ocular toxicity findings as encountered in the mitogen-activated protein kinase inhibitor (MEKi). Both the exact underlying mechanism and the management of ocular toxicity are still debated. We report a case of bilateral pan-FGFR Inhibitor-Associated Retinopathy (FGFRAR) in pulmonary angiosarcoma with FGFR mutation, treated with erdafitinib. To investigate choroidal alterations as a potential pathophysiological mechanism of a serous detachment, the morphology of patient’s choroid and choriocapillaris were assessed. To the best of our knowledge, flow void analysis in macular OCT angiography is reported here for the first time.

## 2. Case Presentation

A 64-year-old Caucasian male patient was referred for ophthalmological examination before initiating a treatment with erdafitinib. He was known to have a history of progressive metastatic pulmonary angiosarcoma and stable, undifferentiated locoregional gastric cardia adenocarcinoma.

The patient started with first-line chemotherapy of Cisplatin, Docetaxel and Fluorouracil 5 (5-FU). Due to the progression of the pulmonary angiosarcoma after three cycles of chemotherapy, he received a second line of chemotherapy with paclitaxel and ramucirumab. No steroids or other medications were prescribed.

The pulmonary angiosarcoma continued to progress over the following months, while the cardia adenocarcinoma remained stable.

Next generation sequencing of the angiosarcoma revealed an exon 14 mutation in FGFR1 (NM023110.2 (FGFR1):c.1966A>G) as the only pathogenic mutation. Due to the dual malignancies, the patient was not eligible for clinical trials with FGFR inhibitors. Therefore, off-label treatment of 8 mg of pan-FGFR inhibitor erdafitinib, once daily, was requested (Novartis), obtained and initiated.

## 3. Results

### 3.1. Ophthalmological General Findings

At the baseline, the patient reported no ophthalmologic complaints.

The best-corrected visual acuity was 20/20 in both eyes. An Amsler grid showed no metamorphopsia. Clear diopters were found upon slit lamp microscopy of the anterior segment. Dilated fundoscopy showed no fundus oculi alterations. Macular optical coherence tomography (OCT) (Figure 1a,b) and angiography (OCT-A) were normal.

Four weeks later, the patient presented with acute photophobia and mild blurred vision in both eyes.

The ophthalmological findings showed bilateral decreased best-corrected visual acuity to 20/25. An absence of the normal foveal reflex was noticed upon fundoscopy. A macular OCT showed bilateral serous neuro-retinal detachment (Figure 1c,d). Fluorescein angiography (FA) was performed and revealed no leakage in either eye and no sign of retinal pigmented epithelium (RPE) alteration (Figure 2a,b). Fundus autofluorescence (FAF) imagining was normal (Figure 3a,b).

Based on these complaints and findings, erdafitinib was suspended. After two weeks, the visual complaints resolved. The best-corrected visual acuity improved to 20/20 in each eye. Macular OCT was normalized in both eyes and the erdafitinib was restarted with a reduced dose of 7 mg daily.

Four weeks after restarting, FGFRAR recurred sub-clinically with stable best-corrected visual acuity and mild subretinal fluid at OCT (Figure 1e,f). The 7 mg daily of erdafitinib was then continued with ophthalmological monitoring.

Bilateral FGFRARshows no signs of staining or leakage in fluorescein angiography mid-phase.

Bilateral FGFRARshows normal autofluorescence imagine.

### 3.2. Choroidal and Choriocapillaris Morphology Assessment

At the baseline, at one and at two months follow-up, choroidal subfoveal and parafoveal thickness were measured by macular-OCT, with no significant changes compared to the baseline. Subfoveal and parafoveal measurements were taken (400 µm from foveal center) with SD-OCT, Spectralis, Heidelberg Engineering. The mean value was calculated between the right and left eye for subfoveal thickness, as well as between the nasal and temporal area for parafoveal thickness. The results were as follows: 167.5, 165, and 168.5 µm as the subfoveal mean value, and 169, 165, and 170.25 µm as the parafoveal mean value.

Macular OCT-A showed no alteration of the deep choriocapillaris plexus. The flow void mean values were calculated as follows: 11.64, 13.16 and 12.24 mm^2^, respectively, at the baseline, at one month and at two months after starting erdafitinib. No significant changes were noticed despite the acute onset and recurrence.

The signal void analysis was performed by a method adapted from Spaide et al. The OCT-A images were imported in the open-source software Image J. Automatic local thresholding was performed with the Phansalkar method using a radius of 15 pixels. Particle analysis was executed on the resulting image (Figure 4a,b), leading to a calculation of the total flow void area in mm^2^ [1,2]. Of note, the debate on the visualization of vessels beyond the RPE is not over, and might bring new insights on how projection artefacts could influence choriocapillaris quantification. Therefore, these data should be interpreted with care.

## 4. Discussion

Here we present a case of FGFR-associated retinopathy with a bilateral serous neuro-retinal detachment after erdafitinib. Similar clinical cases have previously been reported in the literature describing macular unifocal or multifocal serous neuroepithelial detachments [3,4,5,6]. To the best of our knowledge, this is the first case documented by choriocapillaris flow void evaluation with OCT-A.

Recently, it has been proven that an increase in ChT with secondary thinning in choriocapillaris and increasing flow void may lead to an accumulation of serosa fluid in pachychoroid spectrum disorders, such as classic serosa centralis retinopathy (CSCR) [7].

CSCR is a choroidal disease in which the capillaries are hyperpermeable, and can lead to a detachment of the overlying retinal pigment epithelium and subretinal fluid. It is typically related to, and associated with, type A personalities, the use of corticosteroids, autoimmune diseases, and sleep disturbance [8]. Arguably, patients with such advanced cancer can be assumed to meet more than one of these risk factors that would predispose them to having CSCR at the baseline, which may be asymptomatic [9]; however, in our case, it was not, as proven by baseline examination.

The prognosis of typical CSCR cases is self-resolution in 2 to 3 months, usually with the recovery of VA within 3 months. However, some data suggests that up to 15% of patients have persistent symptoms with persistent subretinal fluid [10].

Pachychoroid spectrum diseases are characterized by diffuse or focal areas of increased ChT of more than 300 µm, dilated choroidal vessels, and structural changes such as thinned choriocapillaris and Sattler’s layer overlying the pachyvessels [7]. In our case, subfoveal and parafoveal ChT measured by macular OCT appeared normal, and remained stable at FGFRAR onset and relapse.

In addition, choriocapillaris flow void investigated by OCT-A showed no significant changes.

Considering our results, the FGFRAR in this patient does not show the characteristics of a pachychoroid spectrum disorder; rather, an intracellular mechanism may be at work.

Erdafitinib inhibits tyrosine kinase from transmembrane pan-fibroblast growth factor receptors, which results in suppression of the proliferation of sensitive cancer cells.

Basic FGFR is a neurotrophic factor of which the highest expression is in macroglial cells’ nuclei and the RPE [11,12].

At the intracellular level, FGFR activates two intracellular transduction pathways: mitogen-activated protein kinase (MAPK pathway, RAS/RAF/MEK/ERK pathway) and phosphor-Inositide 3-kinase (PI3K pathway, PI3K/AKT/mTOR pathway).

Both pathways were investigated in RPE cells [11,12].

The MAPK pathway plays an essential role in preventing apoptosis, by protecting cells from oxidative stress and repairing them after mechanical or light-induced injury [11,12].

As the MAPK pathway regulates tight junctions between RPE cells with special action on the fluid transport channel aquaporin 1 (AQP1), modulation of this pathway may disrupt normal fluid transport and lead to an accumulation of fluid under the retina [13]. AQP1 probably contributes to efficient trans-epithelial water transport across the RPE, maintains retinal attachment, and prevents subretinal edema [13].

Through the PI3K pathway, which is important in regulating the cell cycle, RPE cells’ survival is improved under stress conditions [12].

MEKi-associated retinopathy (MEKAR) was recently described separately from CSCR [6]. MEKAR is also characterized by a serous (fluid-filled) retinal detachment without choroidal involvement, secondary to MEKi therapy [13,14,15].

Due to FGFR lying upstream of the MAPK pathway, where MEK as the MEKi target is included, the pathophysiology behind FGFRAR is likely similar to MEKAR.

Clinical features of FGFRAR also resemble previously described MEKAR in terms of patient symptoms, bilateralism, localization of foveal serous detachment, normal findings in FA and FAF examination [3,4]. The present normal ChT and stable choriocapillaris flow void fit a common mechanism.

From what has been observed so far, the accumulation of subretinal serum is highly dependent on the FGFRi/MEKi dose used. In fact, clinical alteration quickly normalizes after the interruption of treatment, and manifests as a reduced stadium at lower doses. Therefore, these data suggest dose-dependent toxicity. A cumulative dose toxicity seems less evident but cannot be totally excluded due to short the follow-up.

## 5. Conclusions

This single case report of FGFRAR, as opposed to CSCR, shows no alteration of choroidal blood flow. Thus, as in MEKAR, RPE-dysfunction based on intracellular alteration plays the most important role in FGFRAR pathophysiology [13]. However, we do not exclude subtle differences between FGFRAR and MEKAR due to the additional inhibition of the PI3K pathway by pan-FGFRi, such as erdafitinib.

Additional cases and further studies at a protein and intracellular level are necessary to confirm our hypothesis regarding this new retinopathy associated with FGFR inhibition.

### Take Home Message


This single case report of Pan-FGFR Inhibitor-Associated Retinopathy is characterized by a serous retinal detachment without choroidal and choriocapillaris involvement.Choroidal and choriocapillaris morphology showed not only stable ChT, but also stable choriocapillaris flow void in macular OCT and OCT-A.Pan-FGFR Inhibitor-Associated Retinopathy does not fit the characteristics of a pachychoroid spectrum disorder; therefore, it should be described separately from CSCR.Rather, the intracellular mechanism in the pathophysiology of Pan-FGFR Inhibitor-Associated Retinopathy should be investigated.


## Figures and Tables

**Figure 1 diagnostics-12-00249-f001:**
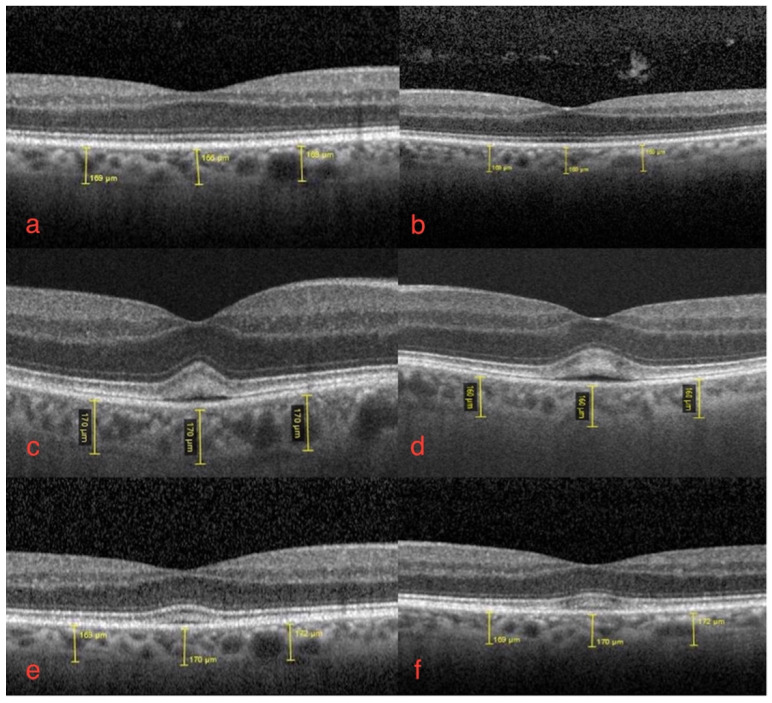
Macular optical coherence tomography of right (**a**,**c**,**e**) and left eye (**b**,**d**,**f**): (**a**,**b**) baseline right and left eye, normal foveal depression and no subretinal fluid; (**c**,**d**) bilateral FGFRAR at one-month follow-up from start of erdafitinib. Foveal serous neuro-epithelial detachment with thickening and high reflectivity of interdigitation zone, disruption of foveal contour with mild increase in central retinal thickening. No sign of increased ChT, no dilated choroidal vessels; (**e**,**f**) bilateral FGFRAR relapse at two-months follow-up from start of therapy: subclinical serous detachments with no change in choriocapillaris thickness. Conserved best-corrected visual acuity.

**Figure 2 diagnostics-12-00249-f002:**
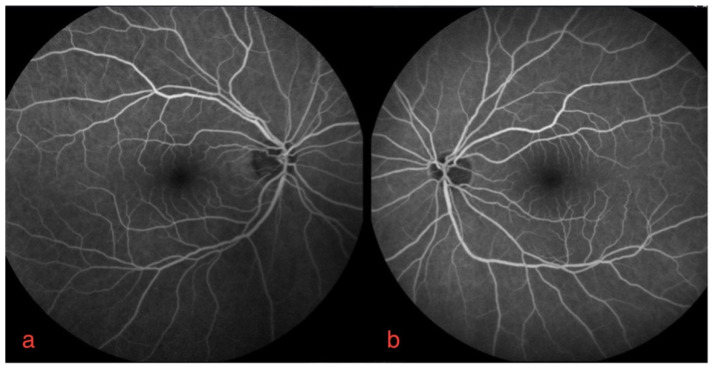
Fluorescein angiography of right (**a**) and left eye (**b**).

**Figure 3 diagnostics-12-00249-f003:**
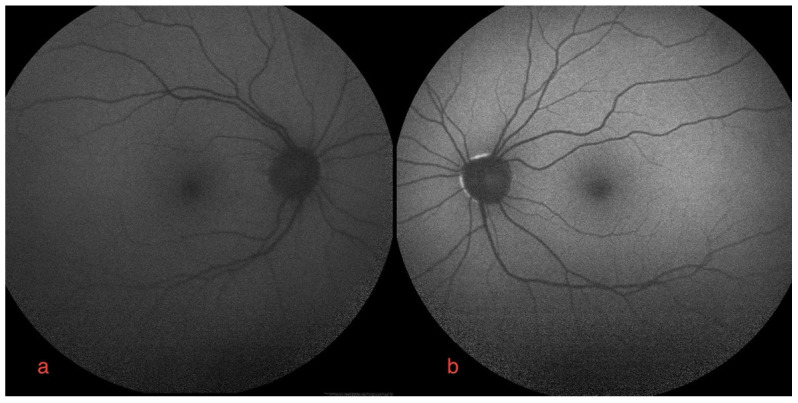
Autofluorescence imaging of right (**a**) and left eye (**b**).

**Figure 4 diagnostics-12-00249-f004:**
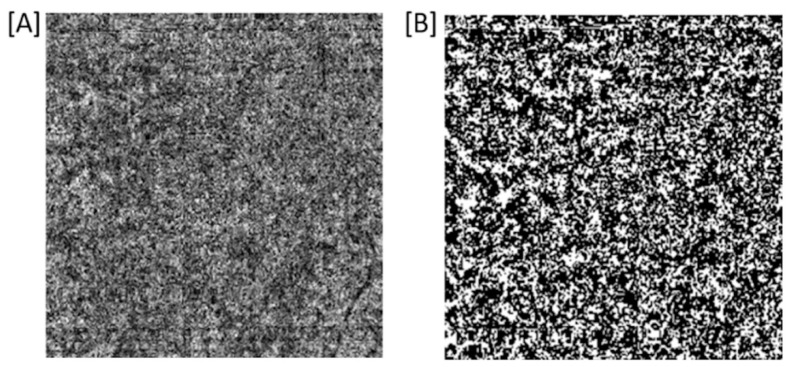
Choriocapillaris flow void of right eye in optical coherence tomography angiography OCT-A CC: en-face optical coherence tomography angiography (SS-OCT, PLEX Elite 9000; Carl Zeiss Meditec, Inc., field of view of 6 × 6 mm centered on fovea) image of the choriocapillaris (CC) layer (**A**) without image processing, and (**B**) after Phansalkar thresholding as used for flow void analysis.

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
