# Peer review of "Choroidal and Choriocapillaris Morphology in Pan-FGFR Inhibitor-Associated Retinopathy: A Case Report"

_diagnostics, 2022, doi:10.3390/diagnostics12020249_

Round 1
Reviewer 1 Report
This is a well-presented case report. Attached is my comment.
This is a well-presented case report that correlates the emerging anticancer pan-FGFR Inhibitor or erdafitinib with development of retinopathy and macular changes. Choroidal thickness and choriocapillaris flow void were respectively measured by macular optical coherence tomography (OCT) and angiography (OCT-A). Data were collected at baseline, at one month and at two months follow-up from start of Erdafitinib in a single case of pulmonary angiosarcoma. Choroidal and choriocapillaris morphology showed stable choroidal thickness and choriocapillaris flow void at FGFRAR onset and relapse. The study concludes that erdafitinib associated retinopathy in this patient includes a bilateral serous neuro-1retinal detachment but was not associated with significant macular alteration by OCT-A and does not seem to match pachychroid spectrum disorder definition.
This case report sets a clinical foundation for further future investigation of the underlying molecular and cellular mechanism for FGFR inhibitor-induced retinopathy.
Author Response
Dear Reviewer 1,
We want to thank you for your professional commentary and for taking the time to read and improve our work.
In the reviewed version you can find some little adjustments of our case report.

Reviewer 2 Report
Please add to the title that is only a case report
Add to the text that the patient wasn´t under any other medication (steroids)
Spaide in written in the text as Spaid
Use most common abbreviation (FA por fluorescein angiography, FAF for fundus autofluorescence)
Choroidal thickness is presented only once as an abbreviation
You cannot conclude that there are not changes in the choroid with only one case. You should point that the choroid was normal in you subject
Author Response
Dear reviewer 2,
Thank you for your kind reply, your time and your contribute to improve our case report.
-Please add to the title that is only a case report
Updated as suggested: line 3
-Add to the text that the patient wasn´t under any other medication (steroids)
Updated as suggested: line 53
-Spaide in written in the text as Spaid
Updated as suggested: line 96
-Use most common abbreviation (FA por fluorescein angiography, FAF for fundus autofluorescence)
Updated as suggested: lines 73,74, 156
-Choroidal thickness is presented only once as an abbreviation
Updated as suggested: lines 18,21,109, 124, 126, 176
-You cannot conclude that there are not changes in the choroid with only one case. You should point that the choroid was normal in you subject
Updated as suggested: lines 165,173

Round 2
Reviewer 2 Report
Authors have changed all the suggestions